# Comparing Plasma Exchange to Escalated Methyl Prednisolone in Refractory Multiple Sclerosis Relapses

**DOI:** 10.3390/jcm9010035

**Published:** 2019-12-22

**Authors:** Steffen Pfeuffer, Leoni Rolfes, Eike Bormann, Cristina Sauerland, Tobias Ruck, Matthias Schilling, Nico Melzer, Marcus Brand, Refik Pul, Christoph Kleinschnitz, Heinz Wiendl, Sven G. Meuth

**Affiliations:** 1Neurology Clinic and Institute for Translational Neurology, University of Muenster, 48149 Münster, Germany; leoni.rolfes@ukmuenster.de (L.R.); tobias.ruck@ukmuenster.de (T.R.); matthias.schilling@ukmuenster.de (M.S.); nico.melzer@ukmuenster.de (N.M.); heinz.wiendl@ukmuenster.de (H.W.); sven.meuth@ukmuenster.de (S.G.M.); 2Institute of Biostatistics and Clinical Research, University of Muenster, 48149 Münster, Germany; eike.bormann@ukmuenster.de (E.B.); cristina.sauerland@ukmuenster.de (C.S.); 3Department of Internal Medicine D, University of Muenster 48149 Münster, Germany; marcus.brand@ukmuenster.de; 4Department of Neurology, University Duisburg-Essen, 45147 Essen, Germany; refik.pul@uk-essen.de (R.P.); christoph.kleinschnitz@uk-essen.de (C.K.)

**Keywords:** multiple sclerosis, optic neuritis, plasma exchange, relapse, class IV, steroids

## Abstract

Intravenous methyl prednisolone (IVMPS) represents the standard of care for multiple sclerosis (MS) relapses, but fail to improve symptoms in one quarter of patients. In this regard, apart from extending steroid treatment to a higher dose, therapeutic plasma exchange (TPE) has been recognized as a treatment option. The aim of this retrospective, monocentric study was to investigate the efficacy of TPE versus escalated dosages of IVMPS in refractory MS relapses. An in-depth medical chart review was performed to identify patients from local databases. Relapse recovery was stratified as “good/full”, “average” and “worst/no” according to function score development. In total, 145 patients were analyzed. Good/average/worst recovery at discharge was observed in 60.9%/32.6%/6.5% of TPE versus 15.2%/14.1%/70.7% of IVMPS patients, respectively. A total of 53.5% of IVMPS patients received TPE as rescue treatment and 54.8% then responded satisfactorily. The multivariable odds ratio (OR) for worst/no recovery was 39.01 (95%–CI: 10.41–146.18; *p* ≤ 0.001), favoring administration of TPE as first escalation treatment. The effects were sustained at three-month follow-ups, as OR for further deterioration was 6.48 (95%–CI: 2.48–16.89; *p* ≤ 0.001), favoring TPE. In conclusion, TPE was superior over IVMPS in the amelioration of relapse symptoms at discharge and follow-up. This study provides class IV evidence supporting the administration of TPE as the first escalation treatment to steroid-refractory MS relapses.

## 1. Introduction

The treatment of acute multiple sclerosis (MS) relapses has remained unaltered for decades. The use of high-dose short-term intravenous (methyl-) prednisolone (IVMPS; 500–1000 mg per day for three to five days) is the accepted treatment for relapses [1,2]. Of note, adrenocorticotropic hormone (ACTH) gel is an alternative for patients who do not tolerate corticosteroids. Moreover, although it has been suggested that intravenous immunoglobulins (IVIG) may be a therapeutic option if steroids are contraindicated, two well conducted randomized controlled trials showed that IVIG as an add-on treatment with IVMPS did not confer additional benefit [3,4].

Interestingly, around 25% of patients remain with significant disability 14 days after IVMPS treatment initiation [5]. For these patients, one option is IVMPS treatment escalation (up to 2000 mg daily) for a further three to five days, as recommended by the national guidelines [2,6]. An alternative option is therapeutic plasma exchange (TPE), which has been proven effective in one small randomized trial that showed the superiority of TPE over sham treatment [7]. The effectiveness of TPE has been reported for all demyelinating disorders of the CNS, including optic neuritis (ON), clinically-isolated syndrome (CIS) and relapsing-remitting MS (RRMS) [8,9,10]. Consequently, several guidelines recommend TPE as an adjunctive treatment for increasing the chances of recovery for steroid-refractory relapses [11,12]. However, most studies evaluating TPE lacked an active comparator (such as escalated IVMPS) and comprised heterogeneous treatment regimens. Also, patients with demyelinating diseases other than RRMS were included in the study populations [7,8,9]. Evidence for IVMPS treatment escalation is to a large part based on a single study that compared MRI endpoints but not clinical endpoints [6]. Furthermore, IVMPS treatment escalation exhibited additional, non-genomic effects in animal models [13]. Robust clinical evidence for the currently recommended treatment sequence (initiation treatment with IVMPS, first escalation treatment with IVMPS, second escalation treatment with TPE) is still lacking [6,11,12]. 

We here analyzed patients with acute relapses of RRMS, CIS or isolated ON who were treated with escalated IVMPS, TPE, or a combination of both.

## 2. Experimental Section

### 2.1. Patients

Between January 2013 and December 2017, all of the in-patients in our department were screened. We identified patients diagnosed with RRMS, CIS, or isolated ON, who received a full course of IVMPS (1000 mg daily for five days without an oral taper) as initial treatment (referred to as “initiation treatment” throughout the manuscript). In a second step, we selected patients who received further relapse treatments (referred to as “escalation treatment” throughout the manuscript) and reviewed their medical chart in detail, using a standardized electronic case report form. All patients included in our analysis were hospitalized in our clinic for both the initiation as well as the escalation treatment. 

The inclusion criteria for final analysis were: (i)established diagnosis of RRMS or CIS according to 2017 revised McDonald criteria [14] or optic neuritis in absence of any other infectious or inflammatory disease of the CNS (especially neuromyelitis optica spectrum disorders)(ii)significant relapse with an increase of the Expanded Disability Status Scale (EDSS) score [15] of at least 1.0 in MS/CIS patients or a decrease of the best-corrected visual acuity (VA) in patients with isolated ON in analogy to a decrease of at least 1 according to the visual function system score (FSS) derived from the EDSS, as inclusion criteria for both initiation and escalation treatment(iii)escalation therapy with either 2000 mg methylprednisolone per day for five days, five cycles of therapeutic plasma exchange or a combination thereof following initiation therapy with 1000 mg per day over 5 days(iv)completion of escalation treatment within six weeks from relapse onset

Therapeutic plasma exchange was performed with a COM.TEC cell separator (Fresenius Hemo-Care GmbH, Bad Homburg, Germany). All patients received treatment via central venous catheters every other day, for a total of five sessions. Per session, one plasma volume was processed, while human albumin solution (5%) was used for substitution. The blood flow rates were 50–70 ml per min. All patients underwent regional pre-centrifugal anticoagulation with citrate, followed by post-centrifugal calcium application. In four cases, the treatment-free interval was extended by another day due to excessive hypofibrinogenemia.

Patients with the following criteria were excluded: (i)pregnancy, as determined by pregnancy test(ii)diagnosis of other systemic inflammatory disorders within the observation period(iii)onset of relapse symptoms more than one month prior to initiation treatment with IVMPS(iv)documentation of a secondary progressive disease course within the observation period

For the patients who received more than one escalation treatment within the observation period, we only evaluated the first relapse to avoid preselection bias.

### 2.2. Assessment of Effectiveness

To overcome limitations of the EDSS in depicting acute, relapse-associated disability, we decided to classify our patients into different response categories. For statistical analysis we applied FSS-based stratification as proposed by Conway and colleagues, which stratifies treatment responses based on peak- and recovery-FSS distances into “good/full”, “partial”, or “worst/no” recovery [16]. We show a modified matrix, as previously used, with outcome stratification in Appendix A [17]. The outcomes were evaluated after treatment completion and at follow-up (3 months after discharge).

Relapses were considered as monosymptomatic when Kurtzke’s FSS of the affected system exceeded the other FSS by at least 1 point. Consequently, if this condition was not given, the relapse was regarded as polysymptomatic. In this regard, patients that either showed similar relapse FSSs for pyramidal and cerebellar functions (3 patients) or pyramidal and sensory functions (5 patients), were assigned to their FSS that was EDSS-defining at follow-up. In addition, 4 patients with spinal lesions displayed a similar FSS for bowel and bladder function and pyramidal function. These patients were subjected to the FSS group “pyramidal”, as no patients were identified with bowel and bladder dysfunction as monosymptomatic relapse.

### 2.3. Assessment of Safety

We also screened patients’ medical charts for severe adverse events and graded the identified events according to recommendations made in the “Common Terminology Criteria for Adverse Events”. The CTCAE classification is as follows: I: asymptomatic testing or mild symptoms without necessity for specific intervention; II: local or noninvasive intervention indicated; III: severe, but not immediately life-threatening event, hospitalization or prolongation of hospitalization necessary; IV: life-threatening event; V: death related to event. The study conduct was ethically approved by the local institutional review board of the University of Muenster, Germany (2017-298-f-S).

### 2.4. Statistical Analysis

The continuous variables are presented as median and interquartile range and compared between groups using a Kruskal–Wallis test. The categorical variables are presented as absolute and relative frequencies and compared using Fisher’s exact test.

To evaluate the influence of multiple variables on the occurrence and outcome of serious adverse events, we applied logistic regression. The results are described with odds ratios (OR), the respective 95% confidence intervals (CI), and Wald-test *p*-values. Either “worst or no treatment response following first escalation treatment” or “stable course versus further deterioration at follow-up” or “development of severe adverse events” were used as dependent variables.

All analyses are explorative and should be interpreted accordingly. *p*-values below 0.05 are considered significant; no adjustment for multiple testing was applied. Statistical analysis was conducted with SPSS Version 25 (International Business Machines Corporation (IBM), Armonk, USA). 

### 2.5. Data Availability Statement

Anonymized data will be shared upon request from qualified investigators.

## 3. Results

### 3.1. Patients

Between January 2013 and December 2017, a total of 541 patients received initiation treatment for MS relapses. Of those, 193 (35.7%) patients were admitted for escalation treatment and all had a persistent functional deficit as defined above. A total of 127 (65.8%) patients received a second course of IVMPS as a first escalation treatment, while 66 (34.2%) patients were directly subjected to TPE. For our final analysis we could include a total of 145 patients: 99 out of 127 patients who received a second course of IVMPS, and 46 out of 66 patients who were directly subjected to TPE. Of note, 53 out of 99 (53.5%) patients were subjected to TPE as the second escalation treatment. None of the TPE patients were re-exposed to increased doses of IVMPS (for consort plot see Figure 1).

Baseline characteristics of all treatment groups (IVMPS, TPE, and IVMPS + TPE) are shown in Table 1. Patients who did not receive additional TPE presented with lower peak relapse EDSS (median: IVMPS: 2.0; TPE: 3.0; IVMPS+TPE: 3.0; *p* = 0.003). Otherwise, patient characteristics showed no significant differences. The patients were, on average, young and early in their disease course, with only one patient being above 60 years old. The median time from retrospectively identified disease manifestation to current presentation was 1 year, and for 40% of patients it was their first demyelinating event. 

One hundred and thirty-two patients fulfilled the 2017 revised McDonald criteria for the diagnosis of RRMS at relapse onset, whereas eight patients presented with isolated optic neuritis and five patients fulfilled the criteria for CIS. There were no differences in distribution between escalation treatment groups (*p* = 0.756).

Accordingly, the majority of patients did not receive disease modifying treatment (DMT) at relapse onset (62.1%). The treatment approved for mild to moderate courses of RRMS was administered to 22.8% of patients, whereas 15.2% received substances approved for the treatment of active RRMS (for a detailed description of administered DMT, see Appendix A). The DMT subset use was evenly distributed between groups (*p* = 0.793). In 137 out of 145 patients the relapse was considered monosymptomatic. The most common relapse presentation was optic neuritis (69 patients; 47.6%). Generally, the frequencies of affected functional systems did not differ significantly between treatment groups (*p* = 0.236). Polysymptomatic relapses occurred in eight patients with infratentorial or spinal lesions and were assigned as outlined in the methods, according to their FSS that was EDSS-defining at follow-up.

### 3.2. Immediate Effects of Escalation Treatment

According to the previously described FSS-distance related analysis matrix, 28 (60.9%) patients showed good/full recovery following TPE, while 15 (15.2%) patients showed good/full recovery following escalation treatment with IVMPS. Partial recovery was observed in 12 (32.6%) TPE treated patients and in 15 (15.2%) IVMPS treated patients. Finally, no or worst recovery was documented in three (6.5%) TPE treated patients and in 69 (69.7%) IVMPS treated patients (*p* < 0.001, see Figure 2A). Next, 53 (53.5%) patients underwent rescue therapy with TPE following IVMPS, whereas the other patients received no further treatment prior to discharge irrespective of their response. Precise information on why no further treatment was given was not always available; patients’ refusal of apheresis treatment was documented as reason in at least eight cases.

After the second escalation treatment with TPE, 25 (47.2%) patients showed a full response and 17 (32.1%) patients remitted partially, while 11 (20.7%) patients were unresponsive to the treatment (see Figure 2B). We performed regression analyses in order to evaluate the possible confounders and to check whether the higher proportion of treatment-resistant patients following IVMPS+TPE versus TPE alone was systematically influenced by different factors/confounders. Logistic regression analysis included “sex”, “age”, “affected function system (visual vs. other)”, “disease duration”, “baseline EDSS”, and “time to treatment initiation”. The adjusted odds ratio for “worst/no” treatment response was 39.01 (95%–CI: 10.42–142.71; *p*<0.001), favoring TPE treatment as the first escalation treatment (for full regression model see Appendix A).

### 3.3. Sustained Effects of Escalation Treatment

Most patients were revisited at our outpatient clinic three months after discharge in order to re-evaluate the outcomes of relapse treatment and to initiate immunomodulatory treatment. A total of 135 (93.1%) patients were evaluated, with no significant differences between groups in terms of attendance (IVMPS: 93.0%; TPE: 90.5%; IVMPS+TPE: 91.8%; *p* = 1.000). The median follow-up duration was 95.5 days (IQR: 86–112), with again no relevant differences between treatment groups (*p* = 0.379). Eight patients reported further relapses with symptoms distinct from previous ones (6 patients/IVMPS group, one patient/TPE group, and one patient/IVMPS+TPE group); and three of these relapses affected the same functional system (optic nerve: two; brainstem: one; onset 53, 64, and 82 days after discharge, respectively). After excluding these patients, we re-evaluated the FSS according to the Conway model. In the IVMPS group, we found a significantly larger proportion of deteriorating patients (41.9%; vs. 12.2% for IVMPS+TPE and 7.1% for TPE; *p* = 0.001). The multivariable odds ratio for further deterioration of relapse symptoms at follow-up was 6.65, favoring the conduction of TPE (95%–CI: 2.52–17.54; *p*<0.001; for full regression model see Appendix A).

### 3.4. Safety 

Out of 145 patients, 116 (80.0%) experienced at least one single adverse event (Table 2). IVMPS treatment was frequently associated with hypertension, hyperglycemia, and hypokalemia, making it necessary to regularly substitute potassium (orally). Temporary insulin treatment was necessary in 14 patients (IVMPS: 6; IVMPS+TPE: 8; TPE: 0). Conversely, coagulopathy was associated with apheresis treatment. However, those events were mostly considered °II according to CTCAE. Infections were observed more often in patients who received two courses of IVMPS and among those, four were considered CTCAE °III due to the prolongation of hospitalization. In particular, one case of central venous catheter-associated septicemia required 14 days of vancomycin treatment until full recovery. Hypotension and coagulopathy each resulted in at least one treatment interruption in 28 TPE treated patients, whereas treatment interruption due to hypertension occurred in two IVMPS treated patients (systolic blood pressure>180mmHg each). Notably, we observed thromboembolic events in four out of the 99 patients exposed to escalated IVMPS, including one case of cerebral venous sinus thrombosis.

We evaluated whether the amount of previously administered IVMPS (initiation treatment with IVMPS and first escalation treatment with TPE vs. initiation and first escalation treatment with IVMPS and second escalation treatment with TPE) influenced the risk for serious adverse events (defined as CTCAE °III) during TPE treatment. The model resulted in an adjusted odds ratio of 4.63, favoring early treatment with TPE (95%–CI: 1.35–15.91; *p* = 0.015). However, severe adverse events were also more abundant in patients with longer disease duration, higher baseline EDSS, or longer time to treatment initiation (for full regression model see Appendix A).

## 4. Discussion

Several studies have documented the beneficial effects of TPE treatment in acute relapsing MS, but virtually all study designs suffered from significant limitations. Studies were either one-armed, had varying treatment regimens, or consisted of a heterogeneous study population in terms of age, pre-treatment, disability, and disease subgroups (CIS, RRMS, and ON; but also neuromyelitis optica-spectrum disorders and other non-specified entities of CNS-demyelination) [8,9,18,19]. Moreover, a relevant proportion of studies on apheresis treatment solely included ON patients and only a few studies with diverse RRMS patient populations described the affected function systems or acute lesion localization in detail. Consequently, next to escalated IVMPS, the international guidelines recommend TPE as one option for treatment escalation following relapse, while refraining from recommending a specific treatment sequence [11].

Our retrospective cohort is well-defined and representative of more than 500 MS in-patients treated for acute relapses in our hospital within the past 5 years. These MS patients were young and mostly at the beginning of their symptomatic phase and therefore of special interest. Effective therapeutic interventions in this early phase of MS may positively influence long-term outcomes, as both relapse frequency and residual disability can be significantly impacted [20,21].

In our cohort, early apheresis treatment resulted in significantly higher response rates compared to escalation treatment with IVMPS. Interestingly, patients who underwent two courses of IVMPS prior to TPE showed poorer response at discharge compared to patients who only had one course of IVMPS prior to TPE. One explanation could be the longer time to apheresis treatment when conducted as the second instead of as the first escalation treatment. Notably, previous studies recommended the initiation of apheresis no later than six weeks after relapse onset in order to allow for the maximum efficacy of TPE, and all patients in this study were below this threshold [22,23,24]. We also hypothesize that the restitution of blood–brain barrier function, as induced by excessive doses of corticosteroids, might hamper the drainage of immunoglobulins and further inflammatory factors towards the blood, where they are ultimately cleared by TPE [25]. Ultimately, MS lesion pathology could have differed between patient groups. A so-called “type-2 lesion pattern”, which is defined by the presence of immunoglobulins within MS lesions, was identified as a strong predictor for the success of TPE [24]. However, this information is usually not available in clinical routines and markers that have been supposed to be associated therewith, such as the presence of ring-enhancing lesions, could not be evaluated here, as MRI data were not regularly available.

As revealed by follow-up data three months after discharge, patients who underwent apheresis treatment exhibit a lower risk for further deterioration, which is in accordance with a previous report [22]. A likely explanation is the higher capacity of apheresis treatment to stop neuroinflammation and consecutive neuroaxonal degeneration, while IVMPS reverts the conduction block but fails to prevent nerve cell death [26]. This hypothesis is supported by the higher frequency of new relapses in patients who did not receive apheresis treatment, although in the short- and mid-term IVMPS treatment has been associated with a reduction in relapse frequency [27]. However, treatment outcomes after three months were supposed to be representative of long-term residuals, as further recovery was less likely beyond this time point in previous studies [28].

In terms of safety, there are some disadvantages of combining escalated IVMPS and TPE. Patients are exposed to high doses of IVMPS, including all the possible side- effects, without having a demonstrable benefit compared to TPE treatment alone. In line with this; we observed a significant increase of complications for the IVMPS escalation group, including several serious adverse events such as thromboembolism or severe infections.

As is typical for retrospective analyses, potentially unknown confounders that might have guided treatment decisions, such as the personal preferences of the treating consultant as well as the patient, health behaviors, comorbidity and MRI characteristics, challenge our study. In this context, we are not able to retrospectively address the criteria underlying the decision to treat a patient with TPE directly in the first escalation, rather than with escalated doses of IVMPS. Moreover, we have to deal with several limitations, such as bias from the selection and availability of data, recall bias, choice of relevant outcome and the methods of analysis. Furthermore, we have to be aware of limitations concerning the validity of our findings, as it is likely that adverse events are probably underestimated, since it was not known that this information was going to be of interest.

However, a large number of patients in our cohort experienced their first demyelinating event and we analyzed only the first relapse per patient, even though intra-individual differences in steroid-responsiveness over time have recently been described [29]. Furthermore, the previously known poor response to steroids used for relapse treatment was not documented anywhere in our medical charts.

In summary, our study found particular advantages of TPE over escalated IVMPS in escalation treatment of MS relapses. We recommend the rapid admission of steroid-refractory patients to apheresis treatment without escalated IVMPS treatment and identify the need to prospectively evaluate this approach in a contemporary patient cohort.

## Figures and Tables

**Figure 1 jcm-09-00035-f001:**
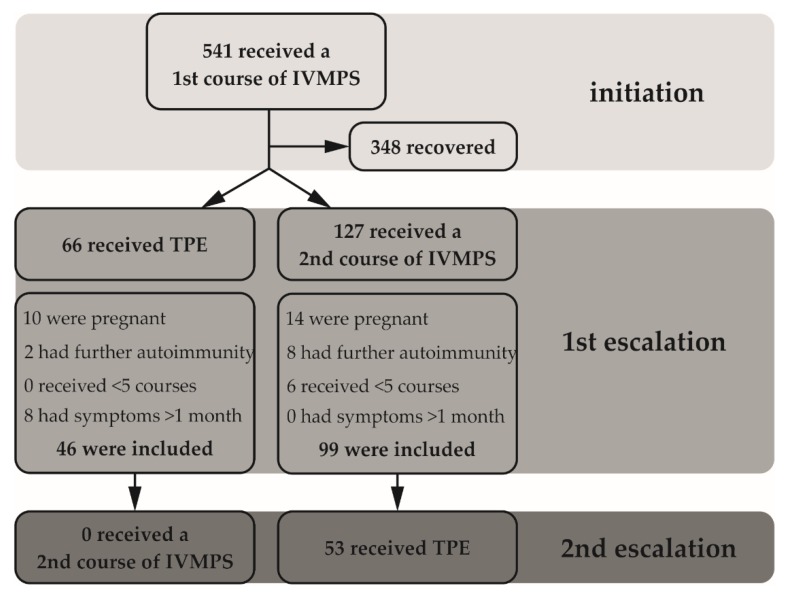
RRMS in-patients who were treated at the study site between January 2013 and December 2017 are described here. The data focus on those patients who received a full course of intravenous methyl prednisolone (5 × 1 g IVMPS) as the first escalation treatment after relapse. Patients who received a lower dosage (e.g., 3 × 1 g IVMPS) were excluded from the primary analysis.

**Figure 2 jcm-09-00035-f002:**
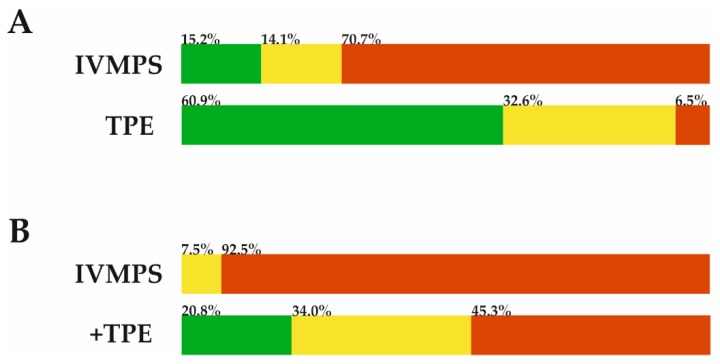
Different response groups following escalation treatment regimens are illustrated (green: good response; yellow: average response; red: worst response). **(A)** Upper bar represents patients who received IVMPS as the first escalation treatment (*n* = 99). Lower bar represents patients who received TPE as the first escalation treatment (*n* = 46). **(B)** Subgroup of patients who received two courses of escalation treatment (*n* = 53). Upper bar shows treatment response after first escalation with IVMPS and lower bar represents results following second escalation with TPE.

**Table 1 jcm-09-00035-t001:** Rescue therapy patient baseline and follow-up characteristics compared between treatment groups.

	TPE	IVMPS	IVMPS+TPE	*p*
**Patients, No.**	*46*	*46*	*53*	-
**Age, yr, median (IQR)**	*33 (29–45)*	*36 (27–43)*	*31.5 (27–41)*	*0.410 **
**Male sex, No. (%)**	*13 (28.3)*	*14 (30.4)*	*14 (26.4)*	*0.922 ^#^*
**MS duration, yr, median (IQR)**				
**- since onset**	*1 (0–3)*	*1 (0–4)*	*1 (0–4)*	*0.574 **
**- since diagnosis**	*0 (0–2)*	*0 (0–2)*	*1 (0–3)*	*0.322 **
**Relapses during last two years, median (IQR)**	*0.5 (0–1)*	*0 (0–1)*	*0 (0–1)*	*0.765 **
**first demyelinating event, No. (%)**	*19 (41.3)*	*20 (43.48)*	*18 (33.96)*	*0.636 ^#^*
**Baseline EDSS, median (IQR)**	*0 (0–1)*	*0 (0–1)*	*0 (0–2)*	*0.397 **
**Relapse EDSS, median (IQR)**	*3 (2–3)*	*2 (2–3)*	*3 (2–3)*	*0.003 **
**Affected function system, No. (%)**				*0.236^#^*
**- visual**	*25 (54.4)*	*25 (47.2)*	*19 (41.3)*	
**- pyramidal**	*4 (8.7)*	*4 (7.6)*	*8 (17.4)*	
**- brainstem**	*8 (17.4)*	*13 (24.5)*	*10 (21.8)*	
**- cerebellar**	*3 (6.5)*	*7 (13.2)*	*1 (2.2)*	
**- sensory**	*6 (13.0)*	*3 (5.7)*	*8 (17.4)*	
**- cerebral**	*0 (0.0)*	*1 (1.9)*	*0 (0.0)*	
**Time to initiation treatment, d, median (IQR)**	*3 (1–7)*	*3 (1–5.25)*	*3 (1–5)*	*0.650**
**Time to escalation treatment, d, median (IQR)**	*12.5 (8.75–16)*	*12 (10–15.25)*	*11 (8.5–14)*	*0.087**

Patient baseline characteristics compared between the different treatment groups. No.: Number; yr.: years; IQR: interquartile range. * Significance levels were calculated using a Kruskal–Wallis test. # Significance levels were calculated using Fisher’s exact test.

**Table 2 jcm-09-00035-t002:** Overview of documented adverse events during hospitalization.

	TPE (*n* = 46)	IVMPS (*n* = 46)	IVMPS+TPE (*n* = 53)
**Hypertension (>135 mmHg SBP)**	1 (2.2)	11 (19.6)	17 (32.1)
**Hyperglycemia (>7.2 mmol/L)**	1 (2.2)	20 (43.5)	32 (60.4)
**Hypokalemia (<3.5 mmol/L)**	4 (8.7)	29 (63.0)	43 (81.1)
**Coagulopathy (aPTT>50 s or INR>1.7)**	16 (34.8)	2 (4.4)	14 (32.1)
**Thrombosis**			
**- Cerebral venous sinus**	-	-	**1 (1.9)**
**- Femoral veins**	-	**1 (2.2)**	**1 (1.9)**
**- Jugular veins/CVC**	-	-	**1 (1.9)**
**Infection**			
**- Thrombophlebitis**	1 (2.2)	3 (6.6)	1 (1.9)
**- Urinary Tract**	4 (8.7)	8 (17.6)	9 (17.0)
**- Respiratory Tract**	-	**2 (4.4)**	**1 (1.9)**
**- CVC infection/septicemia**	-	-	**1 (1.9)**
**(Temporary) treatment interruption**			
**- Coagulopathy**	**4 (8.7)**	-	**7 (13.2)**
**- Hypotension**	**5 (10.9)**	-	**7 (13.2)**
**- Hypertension**	-	**1 (2.2)**	**1 (1.9)**
**- Psychosis**	-	**2 (4.4)**	-
**- CVC dislocation**	**1 (2.2)**	-	**2 (3.8)**
**Pneumothorax**	**1 (2.2)**	-	-
**Patients with at least 1 event**	29 (63.0)	38 (82.6)	49 (92.5)

Overview of adverse events documented during hospital stay. Numbers in brackets represent percentages. Numbers in bold indicate CTCAE °III events. TPE: therapeutic plasma exchange, IVMPS: intravenous (methyl-) prednisolone, SBP: systolic blood pressure; aPTT: activated partial thromboplastin time; INR: international normalized ratio; CVC: central venous catheter.

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
