# Peer review of "Comparing Plasma Exchange to Escalated Methyl Prednisolone in Refractory Multiple Sclerosis Relapses"

_jcm, 2019, doi:10.3390/jcm9010035_

Round 1

Reviewer 1 Report

1.  It might be of value to briefly note/comment on ACTH and IVIG for MS relapses (for completeness sake)

2.  Please clarify: initial treatment was not always 5 (not 3) days IV methylprednisolone with no taper? No one was hospitalized for this first treatment.

3.  Exclusion criteria 4 should be secondary progressive MS (not chronic).

4.  Why was no adjustment for multiple testing applied to P values? (page 3)

5.  Please give more information on those who relapsed on DMT therapy. What were they on, and for how long? Did they differ from the other 62%?

6.  Table 1 might be clearer to add a heading "Rescue Therapy" on top, and to put the TPE group first. 

7.  Table 2, hospitalization is better than residency. TPE column should come first. 

Author Response

Dear members of the editorial board,

We hereby submit a revised version of our manuscript “Comparing plasma exchange to escalated methylprednisolone in refractory multiple sclerosis relapses”. We are thankful for the comments made by the reviewers and have modified the manuscript accordingly. Please find a point-by-point reply attached.

Kind regards

Dr. Steffen Pfeuffer & Dr. Leoni Rolfes on behalf of the authors.

Point-by-point reply

Reviewer #1

It might be of value to briefly note/comment on ACTH and IVIG for MS relapses (for completeness sake)

Response: We thank the reviewer for this important comment and we have now added a brief comment on both ACTH and IVIG for MS relapse treatment (see page 2, lines 53-59).

Please clarify: initial treatment was not always 5 (not 3) days IV methylprednisolone with no taper? No one was hospitalized for this first treatment.

Response: We thank the reviewer for this important note and have added the missing information (page 2, lines 82 and 85-86). As outlined in the experimental section, subheading ‘Patients’, all patients included in the analysis received an initiation treatment with 1000 mg IVMPS daily for five days without an oral taper. All patients were hospitalized in our clinic for both, the initiation as well as escalation treatment. We decided to exclude patients who received their initiation treatment at other hospitals or as outpatients in order to reduce unknown confounders and to exclude for differences in treatment regimen.

Exclusion criteria 4 should be secondary progressive MS (not chronic).

Response: We thank the reviewer for this note and have now adjusted the exclusion criteria as requested (page 3, line 113).

Why was no adjustment for multiple testing applied to P values? (page 3)

Response: No adjustment for multiple testing was applied as all analysis are considered explorative and should be interpreted accordingly. This is also stated in the methods chapter (page 4, line 151).

Please give more information on those who relapsed on DMT therapy. What were they on, and for how long? Did they differ from the other 62%?

Response: We are thankful for this comment. We included “ongoing disease-modifying treatment at relapse onset” as independent variable in our regression models (Supplemental Tables 2-4). In none of these models, DMT had a significant impact on the dependent variables. The statistics have been modified in the manuscript accordingly. Furthermore, detailed information on the DMT administration at baseline is now given in Supplemental Table 1.

Table 1 might be clearer to add a heading "Rescue Therapy" on top, and to put the TPE group first. 

Response: We thank the reviewer for this helpful comment and have now performed the changes as requested (see Table 1, page 6).

Table 2, hospitalization is better than residency. TPE column should come first.

Response: We thank the reviewer for this helpful comment and have now performed the changes as requested (see Table 2, page 8).

Reviewer 2 Report

The manuscript by Pfeuffer & Rolfes et al. reports a retrospective study comparing the use of therapeutic plasma exchange (TPE) with/instead escalated dosages of methyl prednisolone (IVMPS) in relapsing-remitting multiple sclerosis (RR-MS), clinically-isolated syndrome (CIS) and optic neuritis (ON) patients that were not responsive to an initial treatment with IVMPS. The manuscript is overall well written and gives insight on the benefits of TPE over IVMPS. However, some concerns, listed below, have arisen during the review of the manuscript.

- Which criteria were considered for treating a patient with TPE directly in the 1st escalation, instead of with IVMPS?

- In the Introduction, the authors criticize that previous works recruited “patients with demyelinating diseases other than RRMS were included in the study populations” (lines 59-60), while for this study they included RR-MS patients, but also CIS and ON patients Did the authors analyzed the RR-MS patients alone? Please, include the frequencies of each type of disease in the cohort.

- Please, correct some typos found in the text (line 50: “first-line treatment [1,2, even”, there is an square bracket lacking; lines 51-52: “escalation (up 51 to 2000mg daily for further three to five days, as recommended by national guidelines [2, 4]”, lacks a bracket).

Author Response

Dear members of the editorial board,

We hereby submit a revised version of our manuscript “Comparing plasma exchange to escalated methylprednisolone in refractory multiple sclerosis relapses”. We are thankful for the comments made by the reviewers and have modified the manuscript accordingly. Please find a point-by-point reply attached.

Kind regards

Dr. Steffen Pfeuffer & Dr. Leoni Rolfes on behalf of the authors.

Reviewer #2

Which criteria were considered for treating a patient with TPE directly in the 1stescalation, instead of with IVMPS?

Response: We thank the reviewer for this important comment and have now discussed this aspect in the study limitations (see page 9, lines 314-316). Due to the study design and the related limitations, we are not able to address all reasons for treatment decisions, retrospectively. However, we have the feeling that the treating consultant’s preference for a respective escalation treatment might have had major influence. Of course, we were unable to finally validate this finding in this retrospective analysis.

In the Introduction, the authors criticize that previous works recruited “patients with demyelinating diseases other than RRMS were included in the study populations” (lines 59-60), while for this study they included RR-MS patients, but also CIS and ON patients Did the authors analyse the RR-MS patients alone? Please, include the frequencies of each type of disease in the cohort.

Response: We thank the reviewer for this important comment and have now addressed this comment (page 4, lines 180-182). In this study we analysed all patients according to the 2017 revised McDonald criteria. In consequence, the majority of patients with the first demyelinating event already fulfilled the criteria for RRMS.

Please, correct some typos found in the text (line 50: “first-line treatment [1,2, even”, there is an square bracket lacking; lines 51-52: “escalation (up 51 to 2000mg daily for further three to five days, as recommended by national guidelines [2, 4]”, lacks a bracket).

Response: We thank the reviewer for this helpful comment. We corrected the outlined typos.